# A Histone Acetyltransferase Inhibitor with Antifungal Activity against CTG clade *Candida* Species

**DOI:** 10.3390/microorganisms7070201

**Published:** 2019-07-15

**Authors:** Michael Tscherner, Karl Kuchler

**Affiliations:** Max Perutz Labs Vienna, Center for Medical Biochemistry, Medical University of Vienna, Dr. Bohr-Gasse 9/2, A-1030 Vienna, Austria

**Keywords:** *Candida*, CTG clade, histone acetyltransferase inhibitor

## Abstract

*Candida* species represent one of the most frequent causes of hospital-acquired infections in immunocompromised patient cohorts. Due to a very limited set of antifungals available and an increasing prevalence of drug resistance, the discovery of novel antifungal targets is essential. Targeting chromatin modifiers as potential antifungal targets has gained attention recently, mainly due to their role in regulating virulence in *Candida* species. Here, we describe a novel activity for the histone acetyltransferase inhibitor Cyclopentylidene-[4-(4-chlorophenyl)thiazol-2-yl)hydrazone (CPTH2) as a specific inhibitor of CTG clade *Candida* species. Furthermore, we show that CPTH2 has fungicidal activity and protects macrophages from *Candida*-mediated death. Thus, this work could provide a starting point for the development of novel antifungals specific to CTG clade *Candida* species.

## 1. Introduction

Fungal pathogens represent a major health thread worldwide, causing both superficial and systemic infections in immunocompromised people, causing about 1.5 million deaths each year [1,2]. *Candida* species represent one of the most common opportunistic fungal pathogens, being the fourth-leading cause of nosocomial bloodstream infections in intensive care units in the United States, with staggering mortality rates [1,3,4,5]. The majority of *Candida* infections are caused by members of the so-called CTG clade, a group of species that translate the CUG codon as serine instead of leucine. This group also includes *Candida albicans*, the most frequent cause of candidemia, being responsible for at least 40% of all cases worldwide [3,6].

Currently, there are only four main classes of clinically relevant drugs available for treatment of fungal infections [1]. Echinocandins are the newest member of this group and represent inhibitors of cell wall glucan biosynthesis. The first echinocandin, caspofungin, was originally approved by the Food and Drug Administration in 2001 [7]. Since then, no new class of antifungals has entered the market, although two new members of the echinocandin class, micafungin and anidulafungin, have been approved. Development of novel antifungals has been limited by the lack of specific targets in these eukaryotic pathogens. Furthermore, existing antifungal therapies are often associated with severe side effects, as well as an increase in antifungal resistance [3,8,9]. Even for the quite successful class of echinocandins, recent reports indicate increasing prevalence of resistant *Candida* species [10,11]. Thus, there is an increasing need for discovery of novel antifungal targets.

There is now increasing evidence that chromatin modifiers, such as histone acetyltransferases (HATs) or histone deacetylases (HDACs), control fungal virulence and could be potential antifungal targets [12]. For several of these modifiers, it has been shown that deletion of the corresponding gene dramatically reduces fungal virulence [13,14,15,16,17]. Furthermore, an inhibitor of the Hos2 HDAC has even entered clinical trials [18]. Thus, inhibitors targeting chromatin modifiers could be interesting candidates for the development of novel antifungals.

In this work we discovered a novel antifungal activity of the HAT inhibitor Cyclopentylidene-[4-(4-chlorophenyl)thiazol-2-yl)hydrazone (CPTH2). This substance has been described as an inhibitor of the Gcn5 HAT [19]. Here, we demonstrate that CPTH2 can specifically inhibit the growth of CTG clade *Candida* species, and this activity is independent of Gcn5 inhibition. Furthermore, CPTH2 treatment protects primary macrophages from being killed by *Candida albicans*. Thus, this work could provide a starting point for the development of novel CTG clade-specific antifungal drugs to expand the limited repertoire of substances currently available.

## 2. Materials and Methods

### 2.1. Strains, Media, Chemicals and Growth Conditions

*Candida* strains were routinely grown in YPD (1% yeast extract, 2% peptone, 2% dextrose) medium at 30 °C with shaking incubation. For plates, 2% agar was added to the medium. RPMI 1640 medium and Dulbecco’s modified Eagle’s medium (DMEM) were purchased from Thermo Scientific (Waltham, MA, USA). HEK293 cells were cultured in DMEM medium supplemented with 10% fetal calf serum (FCS) at 37 °C with 5% CO_2_. CPTH2 (Sigma-Aldrich, St. Louis, MO, USA), garcinol (Sigma-Aldrich, St. Louis, MO, USA), and anacardic acid (VWR, Radnor, PA, USA) stock solutions were prepared in dimethyl sulfoxide (DMSO). MB-3 (Sigma-Aldrich, St. Louis, MO, USA) and caspofungin (Merck) stock solutions were prepared in sterile water. Strains used in this study are listed in Appendix A.

### 2.2. Spot Dilution and Liquid Survival/Growth Inhibition Assays

Spot dilution assays and growth inhibition assays in liquid medium were performed exactly as described previously [20]. For survival assays, overnight cultures were diluted to an OD_600_ of 0.1, corresponding to ~1 × 10^6^ colony forming units (CFU)/mL in YPD at 30 °C, and 10 µM CPTH2 was added to the medium. Aliquots of the cultures were diluted and plated on YPD plates at the indicated time points to determine CFU values and the survival rate normalized to time point 0 was calculated.

### 2.3. Immunoblotting

*C. albicans* overnight cultures were diluted to an OD_600_ of 0.25 in YPD medium and CPTH2 was added at the indicated concentrations. Cells were incubated at 30 °C for 4 h prior to OD_600_ measurement and harvesting. Whole-cell extract preparation and immunoblotting was performed as described previously [20]. Histone acetylation was detected using an anti-acetyl lysine antibody (NEB, Ipswich, MA, USA), and an anti-PSTAIRE (recognizing Cdc28) antibody (Santa Cruz Biotechnology, Dallas, TX, USA) was used as the loading control. Chicken core histones (1 µg, Millipore, Burlington, MA, USA) were used as positive controls and recombinant histone H3 and H4 (NEB, Ipswich, MA, USA) were used as negative controls (0.1 µg each per lane).

### 2.4. XTT Viability Assay

To determine cytotoxicity of CPTH2 on mammalian cells, 5 × 10^4^ HEK293 cells were seeded into clear flat-bottom 96-well cell culture plates and incubated with the indicated drug concentrations for 24 h. Subsequently, culture supernatants were removed and 0.2 mg/mL XTT sodium salt (2,3-Bis(2-methoxy-4-nitro-5-sulfophenyl)-2H-tetrazolium-5-carboxanilide inner salt, Sigma-Aldrich, St. Louis, MO, USA) and 0.002 mg/mL phenazine methosulfate (PMS, Sigma-Aldrich, St. Louis, MO, USA) in DMEM medium with 10% FCS were added to the cells. After 2 h incubation at 37 °C with 5% CO_2_, absorbance at 490 nm was measured on a Victor^2^ plate reader (PerkinElmer, Waltham, MA, USA). Culture medium was used as blank, and viability relative to untreated cells was calculated. Samples treated with 1% Triton X-100 served as negative controls.

### 2.5. Macrophage Survival Assay

Primary murine macrophages were isolated and cultured as described previously [21]. For interactions with *C. albicans*, 2.5 × 10^4^ macrophages were seeded into 96-well cell culture plates and incubated at 37 °C with 5% CO_2_ for 1 day. Subsequently, 5 × 10^4^ logarithmically growing *C. albicans* cells were washed in phosphate-buffered saline (PBS) and added to the macrophages together with the indicated concentrations of CPTH2. After incubation for 18 h at 37 °C with 5% CO_2_, culture supernatants were used to quantify release of lactate dehydrogenase (LDH) using the CytoTox 96 Non-Radioactive Cytotoxicity Assay (Promega, Madison, WI, USA) according to the manufacturer’s instructions. Absorbance at 490 nm was measured on a Victor^2^ plate reader (PerkinElmer, Waltham, MA, USA). Control samples without CPTH2 treatment were set to 100%.

## 3. Results

### 3.1. CPTH2 Inhibits C. albicans Growth

Lack of the histone acetyltransferase Hat1 increases sensitivity to DNA-damaging agents, as well as to caspofungin, and causes severe growth and virulence defects in *C. albicans* [13,20]. Thus, chemical inhibition of Hat1 could be beneficial for the host. Therefore, we aimed to test a set of known histone acetyltransferase inhibitors for their ability to inhibit Hat1. Since cells lacking Hat1 are highly sensitive to DNA damaging agents, we used sensitivity to the double-strand break-inducing agent methyl methanesulfonate (MMS) as a readout for Hat1 inhibition. We tested wild-type and *hat1*Δ/Δ strains for sensitivity to MMS in the presence and absence of HAT inhibitors using a spot dilution assay. We used a MMS concentration that allows wild-type growth but inhibits the *hat1*Δ/Δ strain. Furthermore, inhibition of Hat1 should already lead to a reduced growth rate in the absence of any additional treatment [20]. We could not observe any synergistic effect for the three HAT inhibitors anacardic acid, MB-3, and garcinol with MMS (Figure 1a). In addition, none of these inhibitors alone had an obvious inhibitory effect on *C. albicans* at the tested concentrations. Surprisingly, however, growth of *C. albicans* was efficiently inhibited by the addition of CPTH2 to the medium at 200 µM, as well as at 50 µM (Figure 1a,b). Lowering the CPTH2 concentration to 25 µM allowed some growth, and deletion of *HAT1* did not affect the sensitivity to this drug (Figure 1c). Importantly, we could also confirm growth inhibition by CPTH2 in liquid YPD medium with an IC_50_ of 3.0 µM for the wild-type strain (Figure 1d). Furthermore, the activity of CPTH2 is not dependent on the culture medium used, since we also observed efficient growth inhibition in RPMI medium (Figure 1e).

Next, we aimed to investigate the mode of action of this drug. To determine whether CPTH2 acts as a fungistatic or a fungicidal, we treated cells in liquid culture with 10 µM CPTH2 and determined CFUs by plating after 3 and 6 h, respectively. No significant growth of *C. albicans* was observed under these conditions (Figure 1f). Importantly, treatment with 10 µM CPTH2 caused a pronounced, time-dependent reduction in viability over a time course of 6 h (Figure 1g). Thus, our data indicate that CPTH2 is a potent growth inhibitor of *C. albicans* with fungicidal activity.

### 3.2. Growth Inhibition by CPTH2 is Gcn5 Independent

Since CPTH2 is described as an inhibitor of Gcn5 [19], we used a *gcn5*Δ/Δ strain to determine its sensitivity to CPTH2. Interestingly, lack of Gcn5 markedly increased the sensitivity to CPTH2 on solid as well as in liquid medium, indicating that inhibition of Gcn5 alone is not the primary cause of its antifungal activity (Figure 2a,b). Deletion of *GCN5* in *C. albicans* causes a dramatic reduction in histone H3 acetylation [15]. To determine if CPTH2 was able to inhibit Gcn5 in *C. albicans* at concentrations affecting fungal growth, we determined the acetylation levels of histones by western blot analysis. However, upon CPTH2 treatment with concentrations up to 10 µM, we could not detect any significant change in the acetylation levels of histone H3 or H4 (Figure 2c,d). As a control we included our *gcn5*Δ/Δ strain and could confirm a dramatic reduction in histone H3 acetylation. Importantly, although we couldn’t detect a significant reduction in histone acetylation, we also observed strong growth inhibition at 10 µM CPTH2 in this experimental setup (Figure 2e). Thus, our data indicate that CPTH2 affects a target independent of Gcn5 at a concentration range that does not inhibit this histone acetyltransferase.

### 3.3. CPTH2 Specifically Inhibits CTG Clade Candida Species

To determine the species specificity of CPTH2-mediated growth inhibition we investigated the sensitivity of different fungal species to this drug. First, we determined the effect of CPTH2 on the growth of *Saccharomyces cerevisiae* and *Candida glabrata*, both of which are distantly related to *C. albicans*. However, both species were not affected by CPTH2 at a concentration strongly inhibiting growth of *C. albicans* (Figure 3a). Next, we tested clinical isolates of different fungal species including members of the CTG clade, a group of closely related species including *C. albicans*, which translate the CUG codon as serine instead of leucine [6]. Interestingly, CPTH2 was able to efficiently inhibit the growth of almost all CTG clade members (Figure 3b). Only *Candida lusitaniae*, and to some extent *Candida tropicalis*, were able to grow, albeit poorly, at 50 µM CPTH2. On the other hand, the non-CTG clade members *Candida kefyr* and *Candida lipolytica* were not affected by this concentration. Interestingly, *Candida krusei* was the only non-CTG clade species showing sensitivity to CPTH2. Furthermore, we observed, to some extent, an inverse correlation between the sensitivity to caspofungin and CPTH2 (Figure 3b). In addition, we could also confirm the sensitivity of the two CTG clade members *Candida dubliniensis* and *Candida parapsilosis* to CPTH2 in liquid medium (Figure 3c). In summary, our data indicate that CPTH2 specifically inhibits the growth of CTG clade *Candida* species.

### 3.4. CPTH2 Protects Macrophages from Candida-Mediated Killing

Finally, we aimed to investigate if CPTH2 can protect host cells from death caused by *C. albicans*. First, we determined the cytotoxicity of CPTH2 to mammalian cells by incubating HEK293 cells with increasing concentrations of CPTH2 and determined viability by using a XTT assay. Even at a concentration of 100 µM, CPTH2 did not have any significant effect on cell viability (Figure 4a).

Next, we incubated primary murine macrophages with fungal cells in the presence or absence of different concentrations of CPTH2. After 18 h incubation, lysis of macrophages was determined by measuring lactate dehydrogenase release into the medium. Strikingly, addition of 10 µM CPTH2 slightly but already significantly reduced macrophage lysis upon interaction with *C. albicans* (Figure 4b). Moreover, increasing the CPTH2 concentration to 25 µM strongly reduced lysis of the immune cells. Thus, our data indicate that CPTH2 can be used to protect host cells from *C. albicans*-mediated death.

## 4. Discussion

In this work, we discovered that the Gcn5 HAT inhibitor CPTH2 can potently inhibit growth of pathogenic fungi and even has fungicidal activity. We have shown that CPTH2 has selective antifungal activity on CTG clade *Candida* species, including *Candida albicans*, one of the most prevalent human fungal pathogens. Furthermore, CPTH2 can protect primary macrophages from *Candida*-mediated death, demonstrating its antifungal activity.

Interestingly, the inhibitory effect of CPTH2 appears to be independent of direct Gcn5 inhibition, since deletion of *GCN5* increases the sensitivity to this drug. Furthermore, histone H3 acetylation is unchanged upon treatment with inhibitory concentrations of CPTH2. These data are in line with previous reports showing that CPTH2 does not only inhibit Gcn5, but also has other targets as well [19,22,23,24]. In *S. cerevisiae* deletion of *GCN5* also increases the susceptibility to CPTH2, albeit at much higher concentrations compared to CTG clade *Candida* species [19,22]. Furthermore, in mammalian systems CPTH2 or its derivatives have been shown to also inhibit the HATs p300 and NAT10 [23,24]. Based on its conserved target site H3K56 and on structural similarities, the fungal-specific Rtt109 HAT has been identified as an orthologue of p300 [25]. In addition, a NAT10 orthologue is also present in *Candida* species [26].

Thus, Rtt109 and Nat10 are additional target candidates for CPTH2, which require further investigation. Notably, all aforementioned HATs are able to acetylate histone H3 [16,24]. Interestingly, the members of the CTG clade, in addition to 2 copies of the canonical histone H3, encode for an additional copy of histone H3, which differs in 3 amino acids [26]. This histone variant is not found in non-CTG clade species, such as *S. cerevisiae* or *Candida glabrata*. Thus, it is tempting to speculate that the CTG clade specificity of CPTH2 could be connected to the presence of this additional H3 variant.

In *S. cerevisiae*, depletion of the HDAC Hda1 can rescue growth defects of a *gcn5*Δ mutant, as well as its increased sensitivity to CPTH2 [22]. Thus, Hda1-related histone modifications might also be important for the antifungal activity of CPTH2 in *Candida* species and might help in identifying its mechanism of action. In addition, in *S. cerevisiae* growth inhibition by 500 µM CPTH2 is more pronounced in medium with low glucose or glycerol [22]. This has been attributed to a regulatory role of the Gcn5-containing Spt-Ada-Gcn5 acetyltransferase (SAGA) complex in respiratory growth. We observed increased sensitivity to CPTH2 in RPMI medium with 0.2% glucose compared to YPD, which contains 2% glucose. Thus, this difference in sensitivity could be due to a conserved role of the SAGA complex in regulating the respiratory metabolism in *C. albicans* and *S. cerevisiae*.

Interestingly, we observe an inverse correlation between CPTH2 and caspofungin sensitivity among clinical isolates of different *Candida* species. Although a larger number isolates have to be screened to strengthen this observation, it raises the interesting possibility of having a complimentary drug or target to echinocandins in cases where resistance is an issue. Thus, our results might provide a starting point for the discovery of novel CTG clade-specific antifungal targets with the potential to reduce side effects and to expand the small number of antifungals currently available.

## Figures and Tables

**Figure 1 microorganisms-07-00201-f001:**
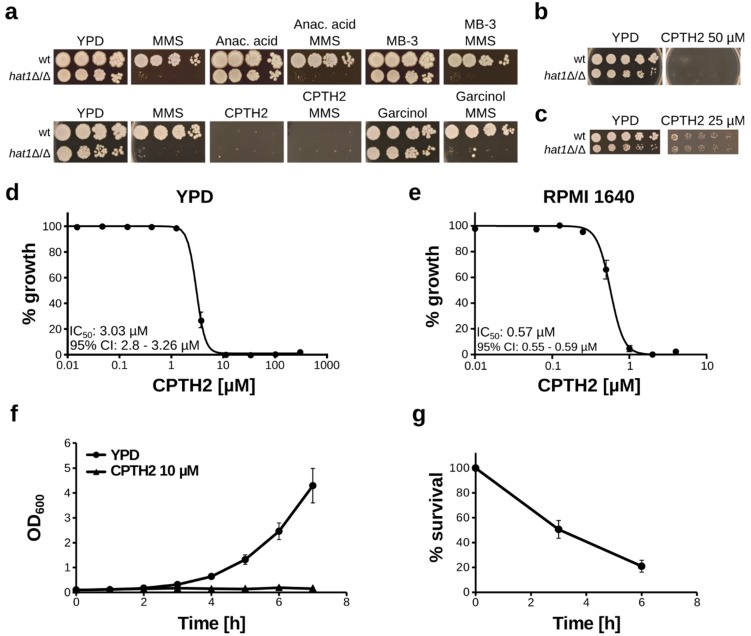
Cyclopentylidene-[4-(4-chlorophenyl)thiazol-2-yl)hydrazone (CPTH2) inhibits *C. albicans* growth. (**a**) Spot dilution assays of *C. albicans* wt and *hat1*Δ/Δ strains using plates containing the indicated histone acetyltransferase (HAT) inhibitors in the absence or presence of methyl methanesulfonate (MMS). Cells were incubated for 3 days prior to imaging. Concentrations used were 200 µM for CPTH2, anacardic acid, and MB-3, 100 µM for garcinol, and 0.03% for MMS. (**b**,**c**) Spot dilution assays as described above (**a**), with 50 µM and 25 µM CPTH2, respectively. (**d**) Liquid growth inhibition assay in YPD medium. *C. albicans* wt cells were incubated with the indicated concentrations of CPTH2 for 24 h prior to OD_600_ measurement. (**e**) Liquid growth inhibition assay as described above (**d**), using RPMI 1640 medium. Cells were incubated for 24 h prior to OD_600_ measurement. (**f**) Growth curves of *C. albicans* wt cultures in the absence or presence of CPTH2. (**g**) Survival of cells upon CPTH2 treatment. Cells were grown in YPD medium and treated with 10 µM CPTH2 for the indicated time. CFUs were determined by plating and colony counting. (**d**–**g**) Data shown are mean ± sd of three (**e**–**g**) or four (**d**) biological replicates.

**Figure 2 microorganisms-07-00201-f002:**
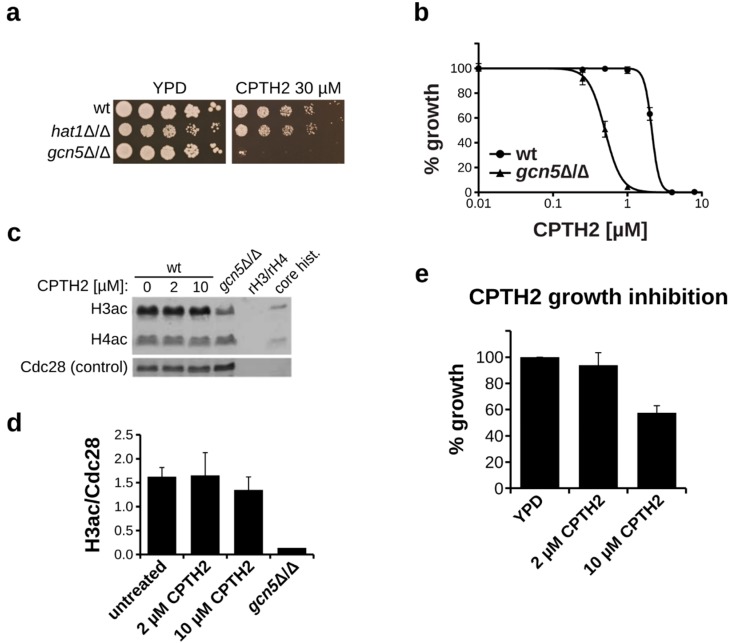
CPTH2 inhibits *C. albicans* independently of Gcn5. (**a**) Spot dilution assay of the indicated *C. albicans* strains on YPD plates. The *gcn5*Δ/Δ strain shows increased sensitivity to CPTH2. (**b**) Liquid growth inhibition assay in YPD medium. *C. albicans* wt and *gcn5*Δ/Δ cells were incubated with the indicated concentrations of CPTH2 for 24 h prior to OD_600_ measurement. (**c**) Immunoblot of histone H3 and H4 acetylation upon treatment with CPTH2. Cells were treated with the indicated concentration of CPTH2 for 4 h prior to SDS-Page analysis followed by immunoblotting with an anti-acetyl lysine antibody. Cdc28 served as loading control. Unacetylated recombinant histones (rH3, rH4) and chicken core histones were used as negative and controls, respectively. (**d**) Quantification of histone H3 acetylation levels upon CPTH2 treatment using immunoblotting as described above (**c**). (**e**) Growth inhibition by CPTH2 of *C. albicans* cultures used for immunoblotting described in (**c**). OD_600_ was measured after 4 hours of treatment. (**b**,**d**,**e**) Data shown are mean ± sd of three biological replicates.

**Figure 3 microorganisms-07-00201-f003:**
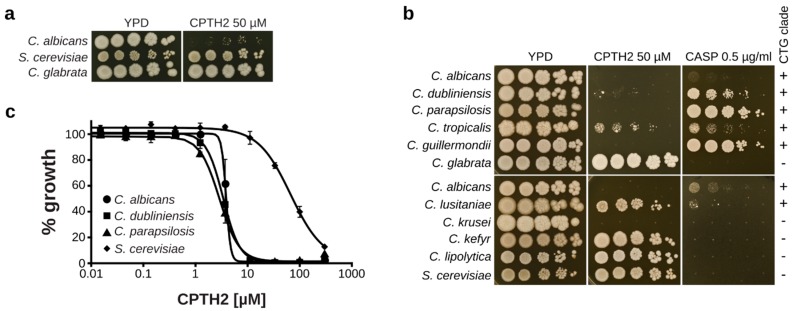
CPTH2 primarily inhibits CTG clade *Candida* species. (**a**) Spot dilution assay of *C. albicans*, *S. cerevisiae*, and *C. glabrata* on YPD plates. Only *C. albicans* shows clear growth inhibition at 50 µM CPTH2. (**b**) Spot dilution assays of different *Candida* species on plates containing either CPTH2 or caspofungin (CASP). CTG clade membership of the respective species is indicated on the right. (**c**) Liquid growth inhibition assay in YPD medium. Fungal strains were incubated with the indicated concentrations of CPTH2 for 24 h prior to OD_600_ measurement. Data shown are mean ± sd of four biological replicates.

**Figure 4 microorganisms-07-00201-f004:**
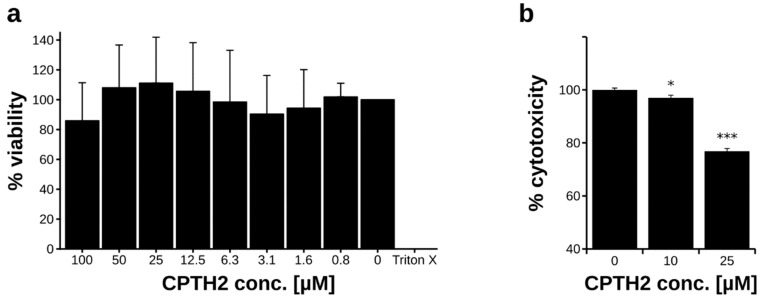
CPTH2 protects macrophages from *C. albicans*-mediated killing. (**a**) Cytotoxicity of CPTH2 was determined using HEK293 cells. Cells were incubated with the indicated concentrations of CPTH2 for 24 hours, followed by quantification of surviving cells using a XTT assay. (**b**) Murine primary macrophages were incubated with *C. albicans* cells and CPTH2 as indicated, and macrophage survival was determined after 18 hours interaction using a lactate dehydrogenase (LDH) release assay. Note: * *p* < 0.05, *** *p* < 0.001. (**a**,**b**) Data shown are mean + sd of three biological replicates.

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
