# Peer review of "A Histone Acetyltransferase Inhibitor with Antifungal Activity against CTG clade Candida Species"

_microorganisms, 2019, doi:10.3390/microorganisms7070201_

Round 1
Reviewer 1 Report
The authors present a concise preliminary study of a potential compound for future therapeutic treatment for Candida species. This is significant since current therapies are not always effective for this important nosocomial pathogen. The study is nice as they look for a compound that would inhibit a Candida protein known to be important for resistance to damaging agents. They demonstrate that CPTH2, a known inhibitor of GCN5 histone acetyl transferase is fungicidal against a subset of Candida species. However, the mechanism of action was not identified, as CPTH2 did not target GCN5 for fungicidal activity. The authors rationally suggest further targets for future work.
1. There is some question regarding the title. The authors state that CPTH2 has clade-specific antifungal activity. However they do demonstrate a few exceptions which are either non-Clade but susceptible or Clade and not susceptible. Albeit this appears to be based on one experiment. It would be curious to know whether there may be some other reason for this distribution.
2. Since this compound is supposed to inhibit histone deacetylation, it is surprising that this is not occurring – can the authors comment? Is this because the concentration is too low – although it appears to be fungicidal in liquid assays. However, the concentration of CPTH2 is lower than any of the spot assays. The inconsistency of CPTH2 concentrations for the different assays sometimes makes it difficult for interpretation. Would it be correct to say that 10µM does not result in histone deacetylation but does kill Candida in liquid culture but does not reduce its ability to kill macrophages?
3. Macrophage killing experiment. The outcome of this study (Fig 4b) is unclear. Were all macrophages killed in the absence of CPTH2? This seems a bit high. Alternatively, was this set as the 100% cytotoxicity mark? Is CPTH2 getting inside the cell? Or is it acting on Candida prior to engulfment?
4. Since it is relatively simple to make knockout strains and/or they are already available, additional studies with knockout strains of the orthologues of NAT10 and Rtt109 would strengthen the manuscript.
5. Normally, revertants of knockout mutant are required for studies. However, in this study it is probably not necessary since the two mutants provide negative results. However, it will be necessary when looking at Rtt109 and NAT10.
Author Response
1.There is some question regarding the title. The authors state that CPTH2 has clade-specific antifungal activity. However they do demonstrate a few exceptions which are either non-Clade but susceptible or Clade and not susceptible. Albeit this appears to be based on one experiment. It would be curious to know whether there may be some other reason for this distribution.
As stated in the manuscript, screening a larger number of strains per species would be required to further strengthen the clade specificity since variation of CPTH2 sensitivity might occur among strains of one species. This could be due to resistance mechanisms like increased efflux, which is a problem for other classes of antifungals.
Thus, we rephrased the title as suggested by the reviewer and removed the word “specific”.
Title in the revised version of the manuscript:
“A histone acetyltransferase inhibitor with antifungal activity against CTG-clade Candida species”
2.Since this compound is supposed to inhibit histone deacetylation, it is surprising that this is not occurring – can the authors comment? Is this because the concentration is too low – although it appears to be fungicidal in liquid assays. However, the concentration of CPTH2 is lower than any of the spot assays. The inconsistency of CPTH2 concentrations for the different assays sometimes makes it difficult for interpretation. Would it be correct to say that 10µM does not result in histone deacetylation but does kill Candida in liquid culture but does not reduce its ability to kill macrophages?
As the reviewer pointed out correctly, we did not see any decrease in the total histone acetylation levels upon treatment with 10 µM CPTH2, although this concentration already inhibited C. albicans growth. However, local histone acetylation at certain genomic regions or at free histones, which represent a very small fraction of the total histone pool, might be inhibited by this compound. This effect would not be visible on a western blot for total histone acetylation since it affects only a minor fraction of the total histones. For example, deletion of the histone acetyltransferase Hat1, which is specific for free histones, does not decrease global acetylation levels.
Thus, treatment with 10 µM CPTH2 does not lead to changes in global histone acetylation, but might inhibit acetylation locally. This could be the reason for the observed growth inhibition at this concentration. Regarding the interaction with macrophages, we could see a rather small decrease in cytotoxicity with 10 µM CPTH2 compared to the clear growth inhibition phenotype in liquid culture. This could be due to sequestration of CPTH2 away from the fungal cells by macrophages. However, further experiments would be required to investigate the interaction of CPTH2 with immune cells, which is beyond the scope of this study.
3.Macrophage killing experiment. The outcome of this study (Fig 4b) is unclear. Were all macrophages killed in the absence of CPTH2? This seems a bit high. Alternatively, was this set as the 100% cytotoxicity mark? Is CPTH2 getting inside the cell? Or is it acting on Candida prior to engulfment?
The untreated control was set to 100%. For clarification, this information was included in the revised version of the manuscript.
As mentioned in the Materials and Methods section, the fungal cells and CPTH2 were added to the immune cells at the same time. Thus, inhibition of Candida can already occur outside of macrophages. Furthermore, since CPTH2 has been used to treat mammalian cells, it seems to be able to enter the cells. Thus, inhibition of fungal cells could also occur upon phagocytosis. However, as mentioned above, further experiments would be required to investigate the interaction of CPTH2 with immune cells.
Text changes in the revised version of the manuscript:
Lines 125-126: “Control samples without CPTH2 treatment were set to 100%.”
4.Since it is relatively simple to make knockout strains and/or they are already available, additional studies with knockout strains of the orthologues of NAT10 and Rtt109 would strengthen the manuscript.
While we agree with the reviewer that testing additional knockouts would be certainly of interest, we think that this would be beyond the scope of this report and subject of future studies.
5.Normally, revertants of knockout mutant are required for studies. However, in this study it is probably not necessary since the two mutants provide negative results. However, it will be necessary when looking at Rtt109 and NAT10.
See point 4 of reviewer 1.
Reviewer 2 Report
Generally, it is a well-written, comprehensive, logical manuscript dealing with the potential anti fungal effect of a histone acetyltransferase inhibitor. I cannot mention any major issues, I have only minor comments to improve the readability of MS:
line 13: write the full name of the discussed histone acetyltransferase inhibitor
line 23: the fourth-leading cause of BSI is true but only in USA and in ICUs, In Europe, there are other epidemiological data
line 30-32: The information is true but the sentence is a little bit misleading. Please reword these sentences and indicate that there are other two approved echinocandins after 2001; micafungin (approved March 2005) and anidulafungin (approved February 2006).
line 61: OD600 of 0.1 correspond to XY CFU/mL
Author Response
Generally, it is a well-written, comprehensive, logical manuscript dealing with the potential anti fungal effect of a histone acetyltransferase inhibitor. I cannot mention any major issues, I have only minor comments to improve the readability of MS:
line 13: write the full name of the discussed histone acetyltransferase inhibitor
The compound name was added as suggested by the reviewer.
Text changes in the revised version of the manuscript:
Line 17: “Cyclopentylidene-[4-(4-chlorophenyl)thiazol-2-yl)hydrazone (CPTH2)”
line 23: the fourth-leading cause of BSI is true but only in USA and in ICUs, In Europe, there are other epidemiological data
The statement was specified as pointed out by the reviewer.
Text changes in the revised version of the manuscript:
Lines 30-31: “… fourth-leading cause of nosocomial bloodstream infections in intensive care units in the US with …“
line 30-32: The information is true but the sentence is a little bit misleading. Please reword these sentences and indicate that there are other two approved echinocandins after 2001; micafungin (approved March 2005) and anidulafungin (approved February 2006).
We rephrased the sentence and included the information about micafungin and anidulafungin as suggested by the reviewer.
Text changes in the revised version of the manuscript:
Lines 42-44:
“Since then no new class of antifungals has entered the market, although two new members of the echinocandin class, micafungin and anidulafungin, have been approved.”
line 61: OD600 of 0.1 correspond to XY CFU/mL
We included the information about the CFU/ml as suggested by the reviewer.
Text changes in the revised version of the manuscript:
Lines 85-86: “… to an OD600 of 0.1 corresponding to ~1 x 106 CFU/ml in YPD …“